# Effect of Al_7_Cu_2_Fe Particles on the Anisotropic Mechanical Properties and Formability of Al-Zn-Mg-Cu-Based Alloy Sheets

**DOI:** 10.3390/ma17235924

**Published:** 2024-12-03

**Authors:** Jonggyu Jeon, Sangjun Lee, Jeheon Jeon, Maru Kang, Heon Kang

**Affiliations:** 1Department of Materials Science and Engineering, Yonsei University, Seoul 03722, Republic of Korea; garuda2927@yonsei.ac.kr (J.J.); or sangjun@iis.u-tokyo.ac.jp (S.L.); 2Institute of Industrial Science, The University of Tokyo, 4-6-1 Komaba, Tokyo 153-8505, Japan; 3Technology Strategy, LG Energy Solution, Daejeon 34129, Republic of Korea; jeheon@lgensol.com; 4Smart Agricultural Machinery R&D Group, Korea Institute of Industrial Technology, Gimje-si 54325, Republic of Korea; mrkang@kitech.re.kr; 5Customized Manufacturing R&D Department, Korea Institute of Industrial Technology, Siheung 15014, Republic of Korea

**Keywords:** Al-Zn-Mg-Cu alloy, Al_7_Cu_2_Fe particles, recrystallization behavior, microstructure, mechanical properties

## Abstract

The presence of Al_7_Cu_2_Fe particles, formed due to Fe impurities in Al-Zn-Mg-Cu alloys, significantly impacts mechanical properties and formability, which is crucial for the use of these alloys in the automotive and aerospace industries. In this study, we prepared Al-Zn-Mg-Cu-based alloy sheets with large (LF), small (SF), and no (NF) Al_7_Cu_2_Fe particles to explore their effects on recrystallization and mechanical behavior. These sheets were fabricated using controlled cooling rates and subsequent thermo-mechanical processing. Fine dispersion of Al_7_Cu_2_Fe particles in the SF sheet led to a substantial reduction in grain size (16.5 μm) and an increase in yield strength (168.6 MPa), albeit with lower ductility (24.6%). In contrast, the NF sheet exhibited a lower yield strength (141.5 MPa) but superior ductility (35.8%) due to the absence of Al_7_Cu_2_Fe particles, which prevented premature fracture. The SF sheet demonstrated lower anisotropy in ductility due to the uniform recrystallized grain orientations, while the LF and NF sheets exhibited significant anisotropy in yield strength. These findings indicate that optimizing Al_7_Cu_2_Fe particle dispersion is key to balancing the strength, ductility, and anisotropy in Al-Zn-Mg-Cu alloys.

## 1. Introduction

Lightweight Al alloys are increasingly regarded as essential materials in the automotive and aerospace industries due to their ability to improve energy efficiency and reduce environmental impact [1,2,3]. In addition to these benefits, Al alloys offer a high strength-to-weight ratio, which enhances vehicle performance by contributing to fuel savings, increased payload capacity, and improved handling. Among the widely used Al alloys, the 5xxx (Al-Mg) and 6xxx (Al-Si-Mg) series are prevalent in automotive sheet applications owing to their excellent formability and corrosion resistance. However, the 7xxx (Al-Zn-Mg-Cu) alloys, while offering superior stiffness and specific strength, have seen limited use in automotive applications, primarily due to their poor formability. This limits their use to parts like front and rear pillars and side impact beams, where high strength is prioritized over ease of manufacturing [4,5]. A comparison between these alloys reveals that while 7xxx alloys provide higher strength, they are costlier and more difficult to process than 5xxx and 6xxx alloys, which are more versatile for forming complex shapes.

The challenge with Al-Zn-Mg-Cu alloys lies in their mechanical limitations, notably their poor formability, which restricts their broader application in automobiles. These alloys are predominantly used in the aerospace sector, where the material can be utilized as thick plates without the need for extensive forming. Therefore, addressing these mechanical deficiencies is crucial for expanding the use of 7xxx alloys in the automotive industry. Improving both their mechanical properties and formability is essential to realize their full potential as a lightweight alternative to steel in critical automotive components.

Over the last several decades, research has focused on the formation behavior of Fe-containing intermetallic (FI) particles and their adverse effects on the mechanical properties of Al alloys. During solidification, coarse FI particles like θ-Al_13_Fe_4_, β-AlFeSi, and Al_7_Cu_2_Fe form due to the low solubility of Fe in solid Al, even though Fe has high solubility in liquid Al [6]. These particles remain undissolved after homogenization at high temperatures above 500 °C [7,8,9,10,11]. In Al-Zn-Mg-Cu alloys, FI particles larger than 1 μm negatively impact the ductility, fatigue resistance, fracture toughness, and corrosion resistance [12,13,14,15,16,17,18,19]. For instance, in automotive crash simulations or high-stress environments, these particles lead to failure modes like void formation and crack propagation, as the brittle FI particles easily fracture, initiating voids that grow and eventually cause material failure [12]. This is particularly relevant in parts subjected to dynamic loads, where the volume fraction of voids increases significantly with increasing Fe content, changing the fracture behavior [14]. For example, fracture toughness in Al-Zn-Mg-Cu alloys decreases from 40 to 30 MPa⋅m^1/2^ as Fe content increases from 0 to 0.6 wt.% [15]. Additionally, Al_7_Cu_2_Fe particles provide preferential sites for fatigue crack initiation and promote pitting corrosion, both of which are critical issues in the automotive and aerospace sectors [16,17,18,19].

Addressing the detrimental effects of FI particles on mechanical properties requires controlling their size, morphology, and distribution. This can be achieved by alloying element addition, cooling rate control, and innovative processing technologies. For example, the introduction of Mn, Sr, and La has been shown to modify the size and shape of β-AlFeSi particles in Al-Si-Mg alloys [20,21,22], while rapid cooling rates can reduce the formation of Al_3_Fe and β-AlFeSi particles [23,24,25,26]. Khalifa et al. [27] demonstrated that ultrasonic melt treatment could convert long, plate-like FI particles into fine, polyhedral particles, dramatically improving their mechanical effects. However, limited research has been conducted on the modification of Al_7_Cu_2_Fe particles in Al-Zn-Mg-Cu alloys through cooling rate control, a gap this study seeks to address.

This study aims to investigate the effects of FI particles on the mechanical properties and formability of Al-Zn-Mg-Cu alloys. By preparing alloys with varying FI particle content—large, small, and absent—and controlling cooling rates (8.2 and 0.55 K/s), we aim to enhance our understanding of how FI particles affect anisotropic mechanical behavior. The microstructural evolution during thermo-mechanical processing, deformation, and fracture will be closely examined to provide insights into improving the performance of these alloys for automotive applications.

## 2. Experimental Procedures

The schematic of the cast alloy manufacturing process is shown in Figure 1a.

The cast alloys were fabricated from two starting materials: Fe-containing 99.8% Al and Fe-free 99.999% Al. Pure Al ingots were placed in a SiC crucible and melted in an electrical resistance furnace at 750 °C. Alloying elements such as Zn (99.8% Zn), Cu (99.9% Cu), Mg (99.8% Mg), Al-15Mn, Al-10Ti, and Al-5Cr (in wt.%) master alloys were added to the melt, and then the melt was held at 750 °C for 2 h. Cast alloys with different cooling rates of 8.2 and 0.55 K/s were prepared by casting the melt into a steel mold preheated to 50 and 400 °C, respectively. The average cooling rates were calculated using K-type thermocouples placed at the center of the mold. The prepared alloys with large FI, small FI, and no FI particles were named LF, SF, and NF alloys, respectively. The chemical composition of the prepared alloys was determined using optical emission spectrometry (SPECTROCHECK, Ametek Co. USA), as shown in Table 1.

The schematic of the alloy sheet manufacturing process is shown in Figure 1b. Homogenization heat treatment was performed at 430 °C for 6 h in an electric box furnace. Hot rolling was performed up to 5 mm at 400 °C and cold rolling was performed up to 1 mm with a reduction of 10% per pass. The cold-rolled sheets were recrystallized at 480 °C for 30 m and then quenched with water.

The microstructure was observed using scanning electron microscopy (SEM, JEOL JSM-7001F, Akishima, Japan) equipped with energy-dispersive X-ray spectroscopy (EDS, AMETEK Octane plus, Berwyn, PA, USA). Electron back-scattered diffraction (EBSD, EDAX-TSL, Japan) was used to obtain the crystallographic orientations of the samples. The samples for the EBSD analysis were mechanically polished, and then the ion milling method was performed to remove the surface strain. The inverse pole figure (IPF) maps of recrystallized and deformed (20%) sheets were obtained at 200× magnification with a step size of 2.0 μm and at 500× magnification with a step size of 1.0 μm, respectively. The orientation distribution function (ODF) and Kernel average misorientation (KAM) map were obtained using the SL-OIM analysis software. The KAM map was calculated using the average misorientation angles of a given point with all its 3rd neighbors.

Uniaxial tensile tests were performed using an Instron-type machine at a strain rate of 10^−3^ s^−1^ at 20 °C. Tensile specimens were machined from the alloy sheets according to the ASTM E8 standard. The mechanical properties and formability of the recrystallized sheets, such as yield strength (YS, 0.2% offset), ultimate tensile strength (UTS), elongation, and r-value, were evaluated for tensile directions of 0, 45, and 90° with respect to the rolling direction (RD). The ex situ tensile tests were conducted at a strain rate of 10^−4^ s^−1^ until fracture, and EBSD analysis was performed at 300× magnification with a step size of 0.7 μm. The specimens were interrupted at strains of 5% and 10% with the goal of investigating the intermediate microstructure.

## 3. Experimental Results

### 3.1. Microstructural Evolution During Thermo-Mechanical Treatment Processes

The SEM images of LF alloy and their corresponding EDS maps are present in Figure 2.

In the EDS analysis, Al, Zn, Mg, Cu, and Fe are indicated by blue, green, red, light green, and orange colors, respectively. The LF as-cast alloy shows that the majority of eutectic particles have components of soluble elements, such as Zn, Mg, and Cu, as shown in Figure 2a. After the homogenization heat treatment, as shown in Figure 2b, most of the eutectic particles containing only Zn and Mg were dissolved in the Al matrix due to their high diffusion rate. Most Al_7_Cu_2_Fe and Al_2_CuMg particles remain undissolved because they are stable even at above 500 °C [9,10,11]. According to a previous report [28], some Al_3_Fe particles formed in the as-cast alloys are transformed into Al_7_Cu_2_Fe particles during high-temperature heat treatment.

The SEM images of as-cast alloys shown in Figure 3a–c.

The as-cast alloys show Al_2_CuMg, MgZn_2_, Al_3_Mg_3_CuZn_2_, Al_3_Mg_2_CuZn_2_, and Al_7_Cu_2_Fe particles, which is consistent with the reported results [29,30]. The SF and NF as-cast alloys show relatively smaller eutectic particle sizes than the LF as-cast alloy due to their fast cooling rate. After the homogenization heat treatment, as shown in Figure 3d–f, the volume fraction of the eutectic particles decreases because the eutectic particles containing Zn and Mg are dissolved in the Al matrix. Since LF and SF homogenized alloys have insoluble Al_7_Cu_2_Fe particles due to Fe impurities (Table 1), they exhibit a relatively larger particle volume fraction than the NF homogenized alloys. The SEM images of as-rolled sheets (Figure 3g–i) show that the eutectic particles are crushed and aligned in a direction parallel to the RD due to the drag effect. The particle distribution of LF, SF, and NF as-rolled sheets is different; the LF sheet has large particles with a large volume fraction, the SF sheet has small particles with a large volume fraction, and the NF sheet has small particles with a small volume fraction.

The IPF maps and ODFs of the sheets recrystallized at 480 °C for 30 m are shown in Figure 4.

The grain boundaries are determined using a 15° misorientation criterion and are represented by solid lines. The average grain sizes of the LF, SF, and NF sheets are 36.9, 16.5, and 81.9 μm, respectively. This difference in recrystallization behavior is determined by the distribution of microscale particles, which will be discussed in Section 4.1. The ODF results (Figure 4d–f) show the texture evolution after recrystallization. The main texture components of the LF sheet can be characterized as cube {001}<100>, copper {112}<111>, and Goss {110}<001> orientations with intensities of 4.298, 2.985, and 2.073, respectively. The texture components of the SF sheet are cube {001}<100>, Goss {110}<001>, P {110}<221>, and copper {112}<111> orientations with intensities of 2.577, 1.460, 2.132, and 1.765, respectively. The texture components of the NF sheet are rotated-cube {001}<110>, inverse brass {112}<110>, and Brass {110}<112> orientations with intensities of 10.110, 6.365, and 4.007, respectively. Interestingly, the cube {001}<100> orientation develops in the LF and SF sheets, whereas the rotated-cube {001}<110> orientation develops strongly in the NF sheet. In general, the recrystallization textures such as the Cube {001}<100>, Brass {110}<112>, and Goss {110}<001> orientations, resulting in an increase in Δr and deterioration of the formability [31].

Figure 5 shows the TEM images and EDS results of LF, SF, and NF recrystallized sheets.

The average particle size of the LF, SF, and NF sheets is 55.3, 51.1, and 44.7 nm, respectively. The particle volume fraction of the LF, SF, and NF sheets is 1.31 vol.%, 0.88 vol.%, and 0.56 vol.%, respectively. The NF sheet exhibited nanoscale particles with relatively low average size and volume fraction due to the small amount of transition metals and the rapid cooling rate. The LF and SF sheets showed Mn-rich and/or Fe-rich particles with rod- and/or round shapes, whereas the NF sheet showed nanoscale AlMgCuZn particles, which is consistent with the reported results [28,32,33,34].

### 3.2. Mechanical Properties and Formability of Recrystallized Sheets

The engineering stress−strain curves of the LF, SF, and NF recrystallized sheets corresponding to the tensile directions of 0, 45, and 90° with respect to the RD are presented in Figure 6.

The average YS for the LF, SF, and NF sheets are 141.7, 151.4, and 130.1 MPa, respectively. The average elongations of the LF, SF, and NF sheets are 24.9, 28.1, and 33.5%, respectively. According to the paper [35], the YS, UTS, and elongation of the Al-6.0Zn-2.5Mg-1.5Cu alloy were approximately 148.8 MPa, 319.1 MPa, and 26.1%, respectively. The relationship between the mechanical properties and microstructural characteristics will be discussed in Section 4.2. The SF sheet exhibits the discontinuous yielding phenomenon (Piobert–Lüders effect) in which solute atoms such as Zn and Mg act as obstacles to the dislocation movement. This phenomenon occurs in SF sheets with a small grain size of 16.5 μm because the mobile dislocations of fine-grained alloys more frequently encounter obstacles than those of coarse-grained alloys [36].

Table 2 summarizes the average grain size, YS, UTS, elongation, work hardening coefficient (*n*), average work hardening coefficient ($\bar{n}$), average plastic strain ratio ($\bar{r}$), and planar anisotropy (Δr) of the LF, SF, and NF recrystallized sheets.

The formability of sheets can be expressed in terms of the Lankford coefficient (*r*-value). It is defined as the ratio of the width strain to the thickness strain. The $\bar{r}$ and Δr are expressed as:(1)$$\bar{r}=\frac{r_{0}+2r_{45}+r_{90}}{4}$$
(2)$$\Delta r=\frac{\left(r_{0}-2r_{45}+r_{90}\right)}{2}$$
where $r_{0}$, $r_{45}$, and $r_{90}$ are the r-values in the 0°, 45°, and 90° directions with respect to the RD.

In all sheets, YS and UTS are relatively high in the 0° direction, and the elongation is relatively low in the 90° direction. In the 0° direction, the SF sheet exhibits a high YS of 168.6 MPa due to grain boundary strengthening, which will be discussed in Section 4.2. The NF sheet exhibits a high elongation of 35.8%. The SF sheet exhibits a high $\bar{n}$ of 0.288 due to the finely dispersed Al_7_Cu_2_Fe particles, whereas the NF sheet exhibits a low $\bar{n}$ of 0.209 due to the absence of Al_7_Cu_2_Fe particles. The LF sheet exhibits a low $\bar{r}$ of 0.69 and a high Δr of −0.13, whereas the NF sheet exhibits a high $\bar{r}$ of 0.83 and a low Δr of −0.05. Comparisons of the anisotropic properties of the YS, elongation, and *r*-value for the LF, SF, and NF recrystallized sheets are presented in Figure 7. In the recrystallized sheets, the YS and elongation are relatively high in the 0° direction, and the r-value is relatively high in the 45° direction. The elongation and *r*-value increase in the order of NF, SF, and LF recrystallized sheets.

### 3.3. Microstructural Evolution During Deformation

The SEM images of the LF, SF, and NF deformed sheets are shown in Figure 8.

After recrystallization heat treatment at 480 °C for 30 m, the majority of eutectic particles containing Zn and Mg are dissolved in the Al matrix, but insoluble Al_7_Cu_2_Fe particles remain undissolved. Thus, there are no eutectic particles in the NF sheet but large and small Al_7_Cu_2_Fe particles in the LF and SF sheets, respectively. The LF and SF deformed sheets show the Al_7_Cu_2_Fe particles cracked perpendicular to the RD and void growth around them. Relatively large voids are formed around Al_7_Cu_2_Fe agglomerates in the LF sheet, whereas finely dispersed voids are formed in the NF sheet.

The IPF and KAM maps of the LF, SF, and NF deformed sheets are shown in Figure 9.

The black points in the IPF and KAM maps correspond to regions where the system cannot specify any orientation, and the confidence index is less than 0.1, indicating highly strained regions or Al_7_Cu_2_Fe particles. Since the KAM analysis only calculates local misorientations less than 5° to eliminate the effect of grain boundaries, higher KAM values indicate a denser accumulation of dislocations [37]. The regions with high KAM values in LF and SF deformed sheets are located around the Al_7_Cu_2_Fe particles (indicated by blue arrows) due to the interaction between the particles and dislocations. The NF sheet has relatively few regions with high strain, even when applying the same strain of 20%, because dislocations can glide easily without obstacles. Some regions with high strain are mainly located around the high-angle grain boundaries. The ODF results in Figure 9g–i show that the cube {001}<100> orientation developed more strongly in the LF and SF sheets after deformation.

The SEM images of the fracture surface of the LF, SF, and NF sheets (Figure 10) show that dimples are developed by nucleation, growth, and coalescence of voids, indicating that ductile failure occurs.

In general, the interfaces between the eutectic particles and Al matrix are strong nucleation sites for the voids because the particle has brittle properties and incoherent interfaces with the Al matrix [38,39]. In the LF and SF sheets, Al_7_Cu_2_Fe particles (indicated by red arrows in Figure 10d,e) are in the center of the dimples, whereas there are no particles in the NF sheet. In addition, some of the coarse Al_7_Cu_2_Fe particles in the LF sheet were broken, whereas the fine Al_7_Cu_2_Fe particles in the SF sheet were not broken. This demonstrates that coarse Al_7_Cu_2_Fe particles can act as strongly preferred sites to develop into micro-cracks through particle fracture or interface separation. Smooth areas in the LF and SF sheets indicate that friction between both surfaces has occurred during the failure process. The SEM image and EDS maps of the fracture surface in the LF sheet (Figure 11) show the Al_7_Cu_2_Fe agglomerates. The Al_7_Cu_2_Fe agglomerates can act as initiation and propagation sites for cracks, forming large-scale necking and then causing early failure.

### 3.4. Mechanical Properties of Naturally Aged Alloy Sheets

The ex-situ tensile test results of the LF, SF, and NF recrystallized sheets naturally aged at 25 °C for 6 months are presented in Figure 12.

The naturally aged LF, SF, and NF sheets exhibit an increased YS of approximately 325 MPa due to the formation of solute clusters and Guinier-Preston (GP) zones [40,41]. GP Zones are formed during natural aging and in the early stages of artificial aging. They act as effective obstacles for dislocations and provide most of the strengthening in Al-Zn-Mg-Cu alloys [42]. Compared to the as-recrystallized state, the naturally aged LF, SF, and NF sheets exhibit differences in strength improvement of approximately 189, 147, and 180 MPa, respectively; however, further study is required to elucidate the reason. The EL of naturally aged LF, SF, and NF sheets is 6.3%, 12.0%, and 17.3%, respectively, which decreases by 74.8%, 57.4%, and 51.1% compared to the as-recrystallized state.

### 3.5. Microstructures of Naturally Aged Alloy Sheets

Figure 13, Figure 14 and Figure 15 show the EBSD results of the LF, SF, and NF sheets during ex-situ tensile tests, respectively.

The regions with high KAM values in LF and SF deformed sheets are located around the Al_7_Cu_2_Fe particles (indicated by red arrows) due to the interaction between the particles and dislocations. However, the NF sheet shows that some regions with high strain are mainly located around the high-angle grain boundaries. Even after natural aging, the NF sheet exhibits the highest elongation, and the LF sheet exhibits the lowest elongation. That is, the reduction in EL after natural aging is significantly alleviated as the Al_7_Cu_2_Fe particles become smaller and disappear (i.e., in the order of the LF, SF, and NF sheets).

## 4. Discussion

### 4.1. Recrystallization Behavior

The average grain sizes of the LF, SF, and NF recrystallized sheets (Figure 4) are 36.9, 16.5, and 81.9 μm, respectively. The average grain size of recrystallized sheets is significantly influenced by particle-stimulated nucleation (PSN). In the PSN phenomenon, the Al matrix around microscale particles, which is highly strained during cold rolling, acts as a preferential nucleation site for recrystallization [43]. The recrystallized grain size is significantly reduced by decreasing the particle size (in a range greater than 1 μm) and increasing their volume fraction due to the frequent occurrence of PSN [44,45]. More specifically, this relationship can be explained by the interaction between microscale particles and dislocations. The particle interspacing (λ) is calculated using the following equation [46]:(3)$$\lambda =\frac{2}{3}d\left(\frac{1}{f}-1\right)$$
where d and f are the size and volume fraction of the particles, respectively. As the particle size decreases and their volume fraction increases, the particle interspacing narrows. The narrow particle interspacing causes strong interactions between the particles and dislocations via the Orowan loops model. The Al matrix around microscale particles acts as a preferred nucleation site for recrystallization due to the strongly accumulated dislocations [47,48]. Consequently, the recrystallized grain size is significantly reduced because the total amount of PSN occurrence increases as the particle size decreases and their volume fraction increases.

In particular, the PSN effect induced by the microscale Al_7_Cu_2_Fe particles is stronger than the effect caused by other eutectic particles. This is because the area of particle deformation zone (PDZ) formed by the Al_7_Cu_2_Fe particles is relatively large due to the local large non-uniform deformation at the tip of rod-shaped Al_7_Cu_2_Fe particles [28]. The number of PSN-induced nuclei increases with increasing area of the PDZ with a high dislocation density and large orientation gradient [49,50]. In addition, the Al_7_Cu_2_Fe particles induce PSN well because they are thermally stable during the thermo-mechanical treatment process. Thus, the size, volume fraction, and distribution of the Al_7_Cu_2_Fe particles in the LF, SF, and NF sheets (Figure 3g–i) have a significant influence on the recrystallization behavior, as shown in Figure 16.

The SF sheet exhibits a finer recrystallized grain size than the LF sheet because the finely dispersed Al_7_Cu_2_Fe particles cause the frequent occurrence of PSN. In the ODF results (Figure 4d–f), the SF sheet has a low maximum intensity of 3.114, indicating the development of a relatively random texture due to the high-efficiency PSN. However, the NF sheet exhibits a relatively large, recrystallized grain size of 81.9 μm because PSN cannot actively occur due to the absence of Al_7_Cu_2_Fe particles.

Micro-scale particles are Al_7_Cu_2_Fe and Al_3_Mg_3_CuZn_2_, which remain even after homogenization heat treatment (Figure 3). Nanoscale particles are particles containing transition metal elements such as Cr, Mn, and Fe and are different from micro-scale particles (Figure 5). Micro-scale particles serve as heterogeneous nucleation sites for recrystallized grains by the PSN during the recrystallization (Figure 16). The micro-scale particles have remained after recrystallization (Figure 8a–c).

The nanoscale particles exist even before the recrystallization process, but they cannot be observed by SEM images because of their small size (approximately 45–55 nm). When Al alloys contain small particles, the recrystallization behavior changes significantly. Finely spaced small particles (<500 nm) inhibit the migration of grain boundaries (known as the Zener pinning effect). Recrystallization is delayed or prevented by the small particles, and then the Al alloys show a large grain size.

### 4.2. Mechanical Properties

The YS in 0° direction for the LF, SF, and NF sheets is 146.4, 168.6, and 141.5 MPa, respectively, as shown in Table 2. The improvement in YS can be explained by the grain boundary and particle strengthening. The relationship between grain size and YS can be expressed in the Hall–Petch equation as:(4)$$\sigma_{Hall–Petch}=\sigma_{0}+k_{y}\cdot d^{-1/2}$$
where d, $\sigma_{0}$, and $k_{y}$ represent the grain size, friction stress, and Hall–Petch slope, respectively. The $\sigma_{0}$ and k*_y_* values reported for the Al-Zn-Mg-Cu alloys are 135 MPa and 120 MPa·μm^1/2^, respectively [51]. When the grain size decreases from 81.9 to 36.9 μm and from 36.9 to 16.5 μm, the grain boundary strengthening is 2.3 and 3.3 MPa, respectively. Based on the Orowan equation, the particle strengthening can be calculated as:(5)$$\sigma_{Particle\;strengthening}=M\frac{0.4Gb}{\pi \sqrt{1-\nu }}\frac{\ln\left(\sqrt{2/3}\bar{d}/b\right)}{\lambda_{p}}$$
where ν, *b*, G, and M are Poisson ratio, Burgers vector, shear modulus, and mean orientation factor, respectively. The ν, *b*, G, and M values for the Al-Zn-Mg-Cu alloys are 0.33, 0.286 nm, 26.9 GPa, and 3.06, respectively [52,53]. $\bar{d}$ is the mean size of the particles, and $\lambda_{p}$ is the mean particle interspacing. The average sizes of the Al_7_Cu_2_Fe particles in the LF and SF sheets are 11.2 and 2.4 μm, and the calculated particle strengthening effect are 3.4 and 13.4 MPa, respectively. Consequently, the particle strengthening of fine Al_7_Cu_2_Fe particles is the main factor contributing to the YS improvement of the SF sheet rather than grain boundary strengthening.

The deformation is accompanied by the movement of dislocations, which interact with obstacles, such as high-angle grain boundaries and eutectic particles. The failure of Al alloys occurs because the stress concentration results in the nucleation, growth, and coalescence of voids, which then develop into the formation of microcracks. Rather than grain boundaries, eutectic particles such as Al_7_Cu_2_Fe with brittle properties and incoherent interfaces with the Al matrix are the strongly preferred sites for the formation of microcracks. The elongations in the 0° direction for the LF, SF, and NF sheets are 24.6, 28.2, and 35.8% (Table 2), respectively. Although the energy required for the formation and propagation of cracks increases with decreasing grain size [54], the NF sheet with a large recrystallized grain size of 81.9 μm exhibits a high elongation. Because the NF sheet has no brittle obstacles, such as Al_7_Cu_2_Fe particles, dislocations glide easily, resulting in a high elongation. On the other hand, in the LF and SF sheets, dislocation accumulation occurs around Al_7_Cu_2_Fe particles, and the particles are cracked perpendicular to the tensile direction, as shown in Figure 8 and Figure 9. These cracks act as strong preferred sites for the nucleation, growth, and coalescence of voids and develop into micro-cracks. Thus, in the LF sheet with Al_7_Cu_2_Fe agglomerates, the Al_7_Cu_2_Fe particles are more likely to crack easily when stress concentration occurs, resulting in an early fracture.

### 4.3. Anisotropic Properties

The anisotropic mechanical properties and formability are influenced by several factors, such as crystallographic texture, morphology, and distribution of second-phase particles, morphology and size of grains, and dislocation arrangement [55,56]. For all sheets, as shown in Table 2, the YS and UTS are highest at 0°, and the r-value is highest at 45°, which is consistent with other results reported for Al-Zn-Mg-Cu alloys [57,58,59]. In the deformed sheets (Figure 8), there are no Al_7_Cu_2_Fe particles in the NF sheet, whereas the LF and SF sheets show large and small Al_7_Cu_2_Fe particles aligned in the RD, respectively. The SF sheet exhibits large anisotropy of YS due to the fine Al_7_Cu_2_Fe particles strongly aligned in the RD. Different particle interspacing (λ) in the 0, 45, and 90° directions with respect to the RD ($\lambda_{0}<\lambda_{45}<\lambda_{90}$) leads to the different strengthening effects. The YS and n (work hardening coefficient) of the SF sheet are particularly high in the 0° direction because the narrow particle interspacing of $\lambda_{0}$ causes a strong interaction between the particles and dislocations. On the other hand, the LF sheet exhibits weak anisotropy of YS owing to the low strengthening effect of large Al_7_Cu_2_Fe particles. Despite the absence of Al_7_Cu_2_Fe particles, the NF sheet with large grains exhibits a large anisotropy of YS due to the development of strong texture with a maximum intensity of 16.058 (Figure 4f). The yielding of materials in which dislocations glide along the slip systems occurs when the resolved shear stress is greater than the critical resolved shear stress. According to Schmid’s law, the critical resolved shear stress depends on the crystallographic orientation, thus, the strong textures cause a large anisotropy of YS [60]. Furthermore, fine grains with various crystallographic orientations reduce the anisotropy of elongation because they operate a similar slip system in all directions [61,62]. Therefore, the LF and SF sheets exhibit low anisotropy of elongation, whereas the NF sheet exhibits relatively high anisotropy of elongation.

Formability is greatly affected by the comprehensive effect of the texture components and their volume fractions in Al alloys [31,63,64]. In general, the brass {110}<112>, P {110}<221>, and Goss {110}<001> orientations exhibit a high $\bar{r}$, but also a high Δr [65]. The rotated-cube {001}<110> orientation exhibits a relatively higher $\bar{r}$ and lower Δr compared to the cube {001}<100> orientation [66]. The NF sheet exhibits a high $\bar{r}$ of 0.83 and a low Δr of −0.05 because the rotated-cube {001}<110> orientation is strongly developed. Since the SF sheet has a relatively low maximum intensity of 3.114 (i.e., the development of random texture), the SF sheet exhibits higher $\bar{r}$ and lower Δr than the LF sheet. This is because finely dispersed Al_7_Cu_2_Fe particles in the SF sheet induce the formation of a random texture through high-efficiency PSN. The LF sheet exhibits a low r-value at 0° due to the development of the strong cube {001}<100> orientation with an intensity of 4.298. In addition, in the LF and SF sheets, the brittle Al_7_Cu_2_Fe particles aligned to the RD adversely affect the formability as they can act as the preferred sites for crack initiation [67].

## 5. Conclusions

This study investigated the effects of Al_7_Cu_2_Fe particles on the anisotropic mechanical properties and formability of Al-Zn-Mg-Cu-based alloy sheets. By varying the particle sizes and distributions, we explored how these factors influence recrystallization behavior, yield strength (YS), ductility, and anisotropy.

The dispersion of Al_7_Cu_2_Fe particles played a significant role in enhancing PSN efficiency, leading to finer grain sizes and improved YS. Fine dispersion of these particles in the SF sheet, for example, resulted in a higher YS while maintaining sufficient ductility, which is critical for materials intended for high-strength applications. These findings underscore the importance of optimizing the particle size and distribution in controlling the mechanical properties of aluminum alloys, making them suitable for high-performance industrial applications, particularly in automotive structural components.

The presence of Al_7_Cu_2_Fe particles also had a notable impact on ductility and microcrack formation. Larger particles in the LF sheet led to early fracture due to void growth around the particles. In contrast, the NF sheet, free from Al_7_Cu_2_Fe particles, exhibited improved ductility, which highlights the potential of developing alloy sheets with tailored particle characteristics to prevent early failure. This insight is crucial for industries like aerospace, where durability and resistance to fracture are paramount.

Anisotropy, particularly in YS and elongation, was significantly influenced by the alignment of the Al_7_Cu_2_Fe particles along the rolling direction. The SF sheet demonstrated lower anisotropy due to the finer particle dispersion, making it more suitable for applications requiring consistent performance across different orientations, such as automotive body panels. In contrast, the NF sheet’s strong texture contributed to higher anisotropy, suggesting that careful control of texture and particle alignment is essential for optimizing anisotropic properties in engineering materials.

This study contributes to a deeper understanding of how particle size and distribution affect the mechanical and anisotropic properties of Al-Zn-Mg-Cu alloys, with direct implications for the automotive and aerospace industries. By refining the control of Al_7_Cu_2_Fe particles, manufacturers can develop materials with enhanced strength, ductility, and formability while minimizing anisotropy. Future research should focus on exploring further alloying strategies and processing techniques to optimize the balance between strength and ductility, with potential applications in lightweight, high-strength components.

## Figures and Tables

**Figure 1 materials-17-05924-f001:**
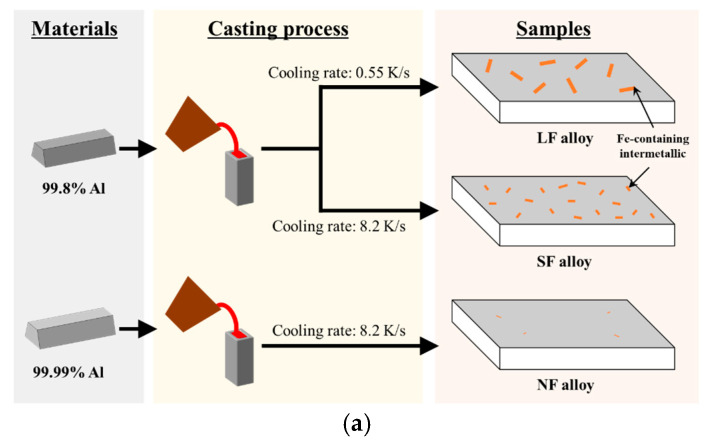
Schematic diagrams of (**a**) cast alloy and (**b**) alloy sheet manufacturing processes.

**Figure 2 materials-17-05924-f002:**
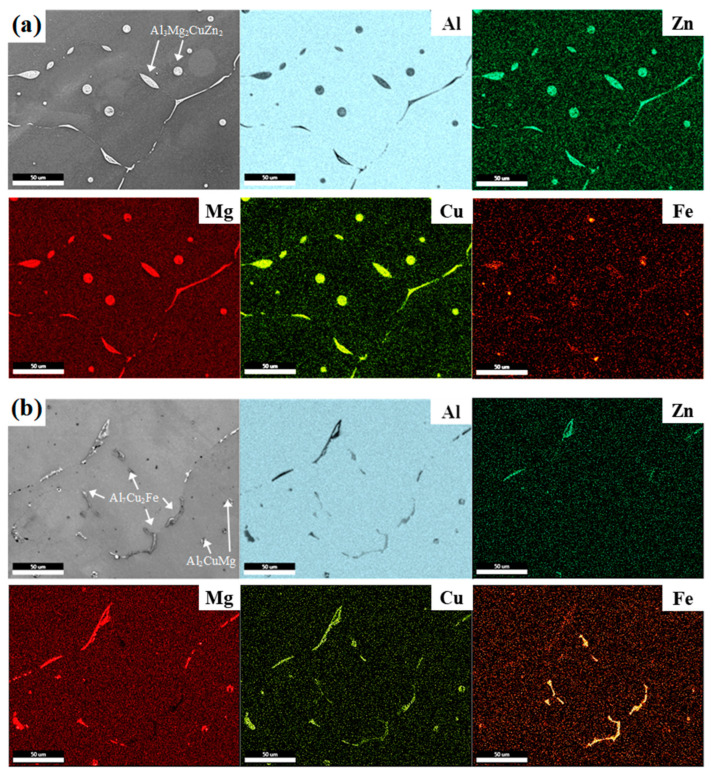
SEM images and EDS maps of (**a**) as-cast LF and (**b**) homogenized LF alloys.

**Figure 3 materials-17-05924-f003:**
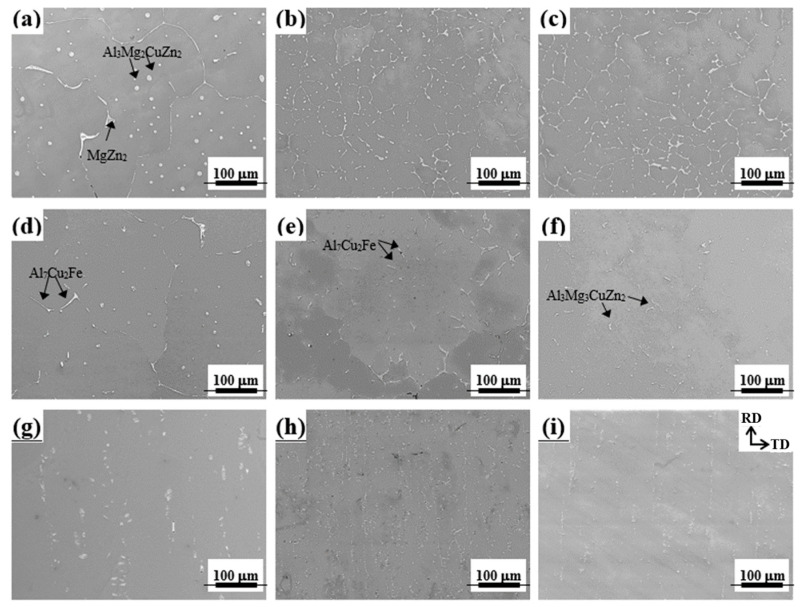
SEM images of the samples under different conditions: as-cast (**a**) LF, (**b**) SF, and (**c**) NF alloys; homogenized (**d**) LF, (**e**) SF, and (**f**) NF alloys; as-rolled (**g**) LF, (**h**) SF, and (**i**) NF sheets.

**Figure 4 materials-17-05924-f004:**
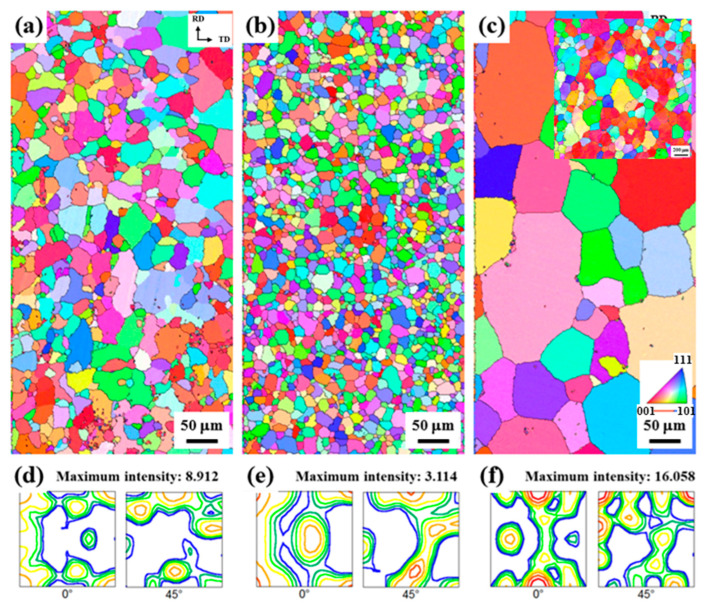
IPF maps and ODFs of the (**a**,**d**) LF, (**b**,**e**) SF, and (**c**,**f**) NF sheets recrystallized at 480 °C for 30 m. Low-magnification IPF map of the NF sheet in Figure 4c.

**Figure 5 materials-17-05924-f005:**
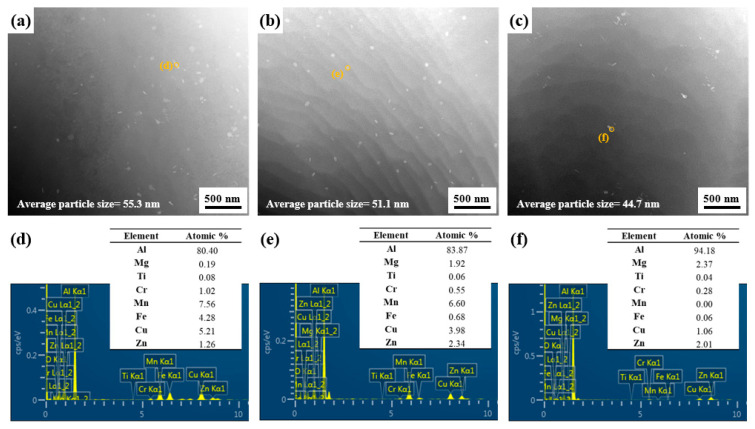
TEM images of (**a**) LF, (**b**) SF, and (**c**) NF sheets recrystallized at 480 °C for 30 m. (**d**–**f**) EDS results of the particles indicated in the figure.

**Figure 6 materials-17-05924-f006:**
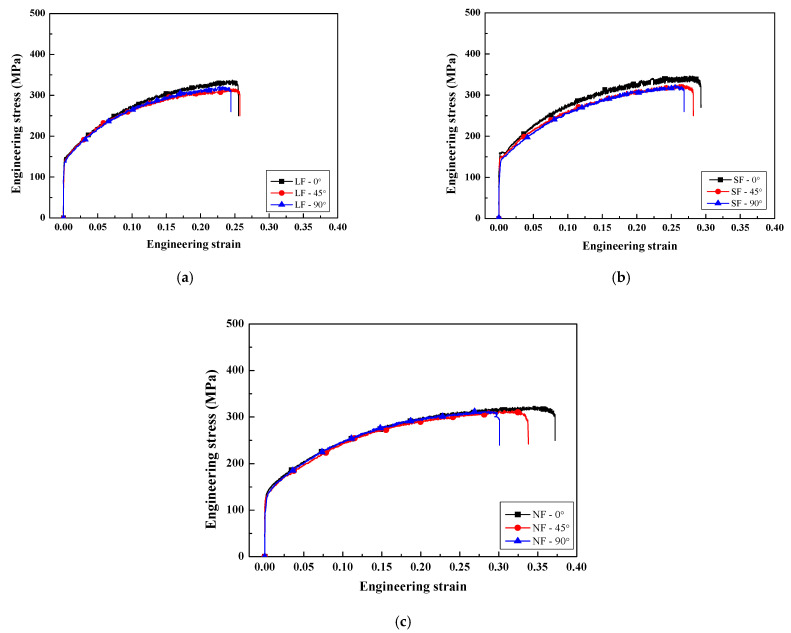
Engineering stress-strain curves of (**a**) LF, (**b**) SF, and (**c**) NF recrystallized sheets corresponding to the tensile directions of 0, 45, 90° with respect to the RD.

**Figure 7 materials-17-05924-f007:**
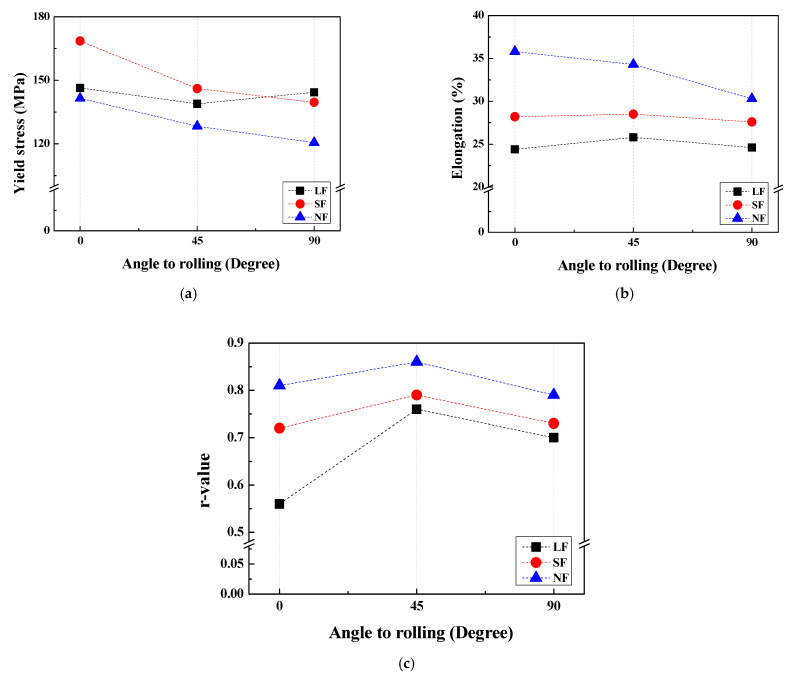
(**a**) YS, (**b**) EL, and (**c**) r-value as a function of the angle to the RD in the recrystallized sheets.

**Figure 8 materials-17-05924-f008:**
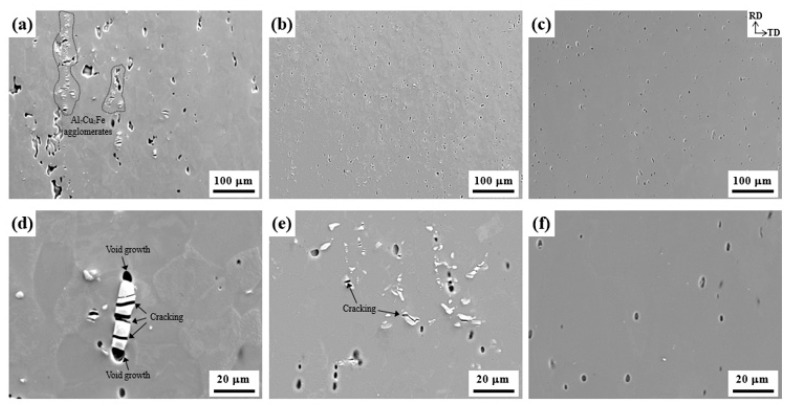
SEM images of (**a**,**d**) LF, (**b**,**e**) SF, and (**c**,**f**) NF sheets deformed by 20%. The tensile direction coincides with the RD.

**Figure 9 materials-17-05924-f009:**
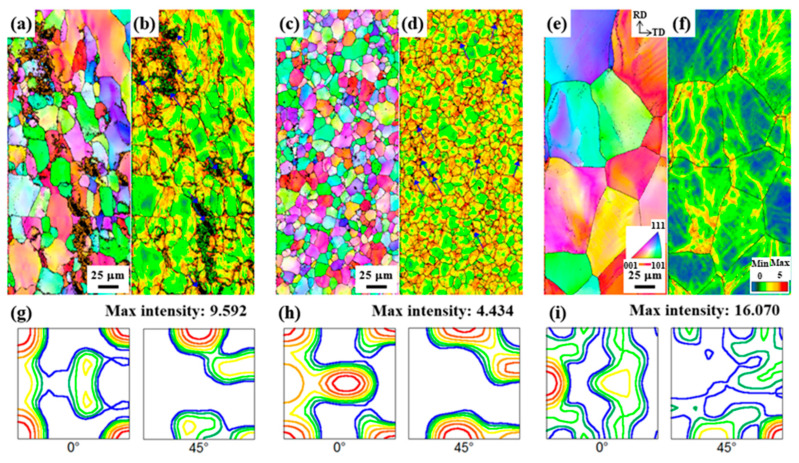
IPF maps, KAM maps, and ODFs of (**a**,**b**,**g**) LF, (**c**,**d**,**h**) SF, and (**e**,**f**,**i**) NF sheets deformed by 20%, respectively. The tensile direction coincides with the RD. The Al_7_Cu_2_Fe particles are indicated by blue arrows.

**Figure 10 materials-17-05924-f010:**
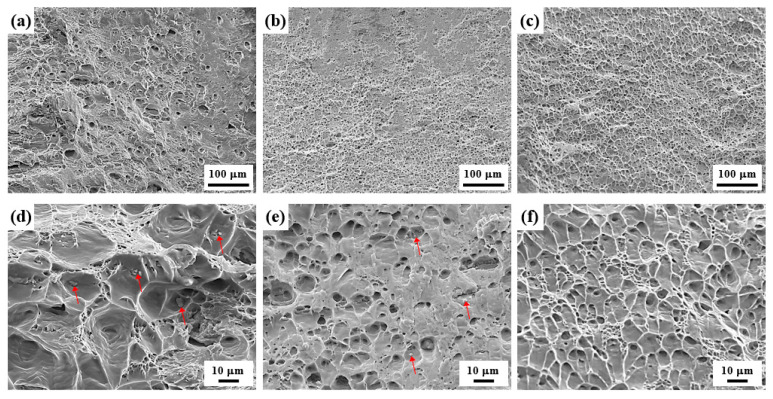
SEM images of the fractured surface of (**a**,**d**) LF, (**b**,**e**) SF, and (**c**,**f**) NF tensile specimens. The Al_7_Cu_2_Fe particles are indicated by red arrows.

**Figure 11 materials-17-05924-f011:**
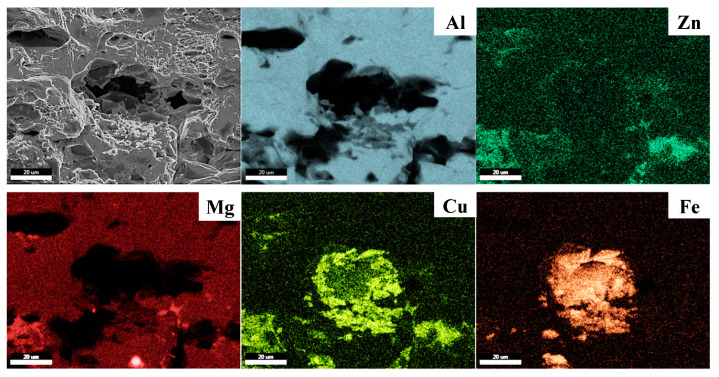
SEM image and EDS maps of the fractured surface in the LF sheet.

**Figure 12 materials-17-05924-f012:**
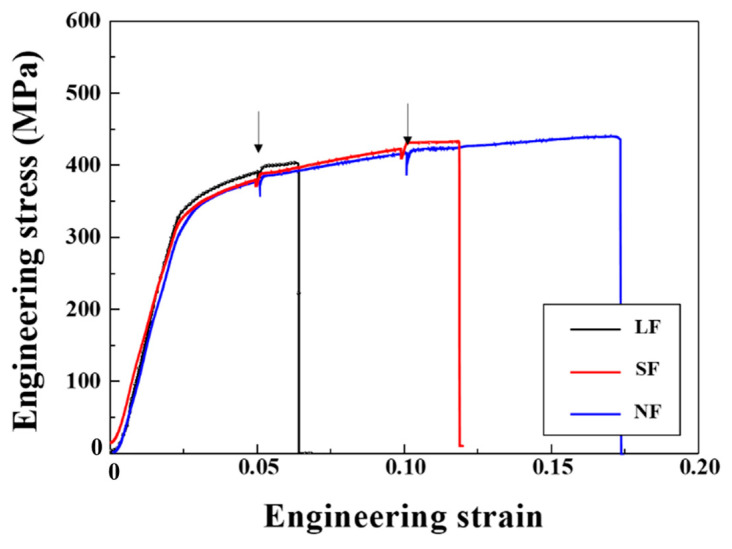
Ex-situ tensile test results of LF, SF, and NF recrystallized sheets naturally aged at 25 °C for 6 months. Tensile tests were performed in the RD. The interrupted 5% and 10% points are indicated by arrows.

**Figure 13 materials-17-05924-f013:**
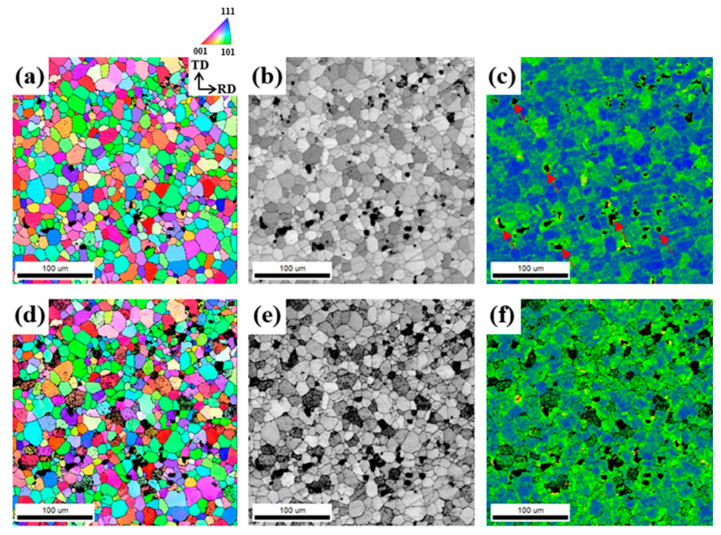
(**a**,**d**) IPF, (**b**,**e**) IQ, and (**c**,**f**) KAM maps of the LF recrystallized sheet naturally aged at 25 °C for 6 months: (**a**–**c**) before deformation and (**d**–**f**) 5% deformed. The Al_7_Cu_2_Fe particles are indicated by red arrows.

**Figure 14 materials-17-05924-f014:**
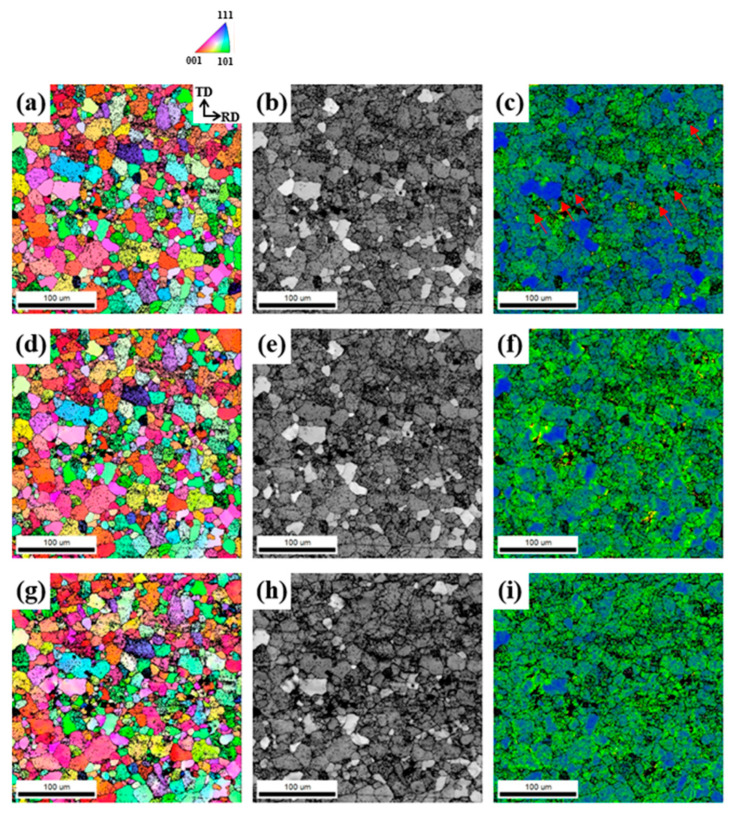
(**a**,**d**,**g**) IPF, (**b**,**e**,**h**) IQ, and (**c**,**f**,**i**) KAM maps of the SF recrystallized sheet naturally aged at 25 °C for 6 months: (**a**–**c**) before deformation, (**d**–**f**) 5%, and (**g**–**i**) 10% deformed. The Al_7_Cu_2_Fe particles are indicated by red arrows.

**Figure 15 materials-17-05924-f015:**
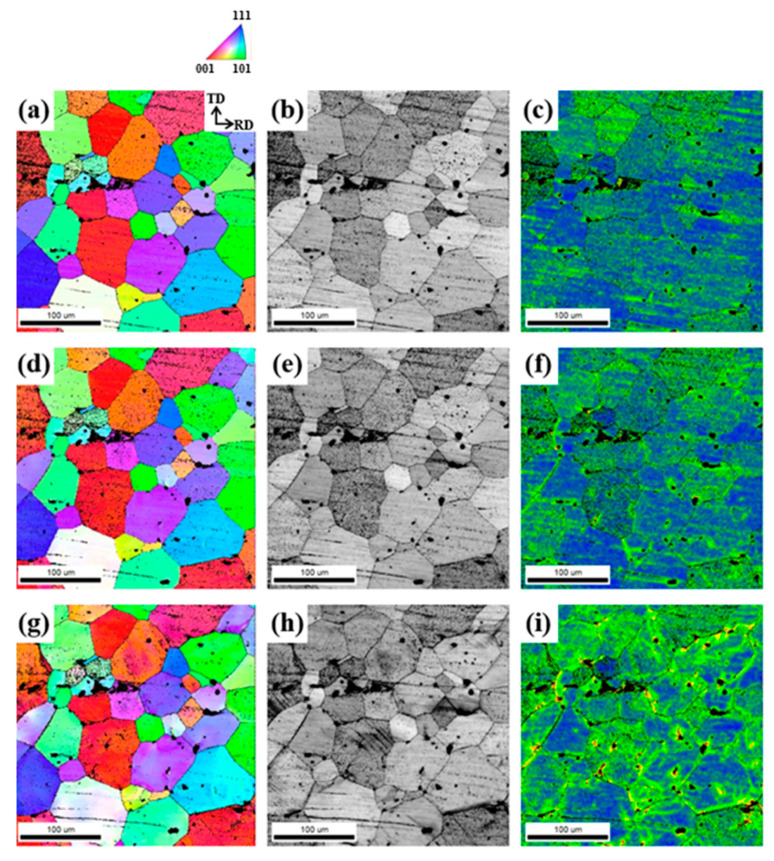
(**a**,**d**,**g**) IPF, (**b**,**e**,**h**) IQ, and (**c**,**f**,**i**) KAM maps of the NF recrystallized sheet naturally aged at 25 °C for 6 months: (**a**–**c**) before deformation, (**d**–**f**) 5%, and (**g**–**i**) 10% deformed.

**Figure 16 materials-17-05924-f016:**
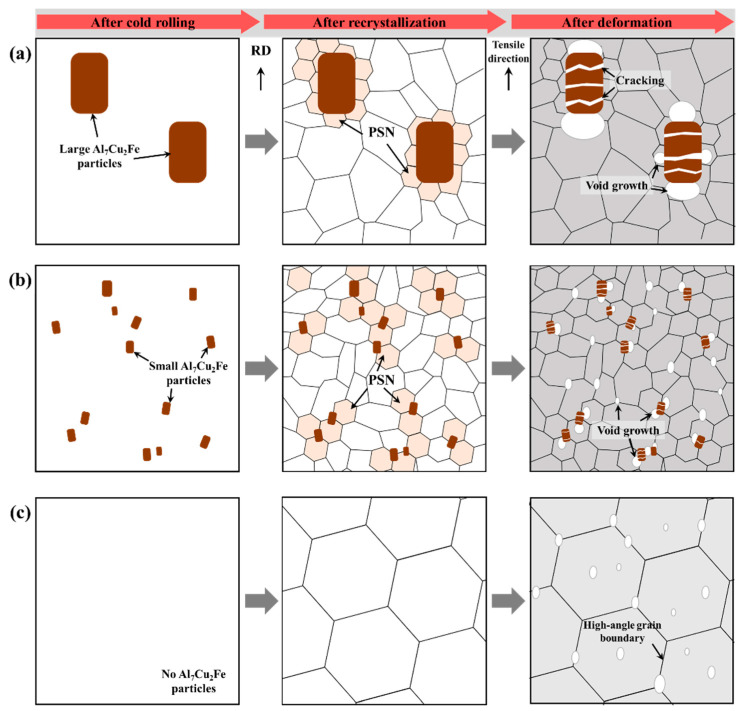
Schematic of the role of Al_7_Cu_2_Fe particles on recrystallization and deformation behavior: (**a**) LF, (**b**) SF, and (**c**) NF alloy sheets.

**Table 1 materials-17-05924-t001:** Measured chemical composition (in wt.%) of the Al-Zn-Mg-Cu alloy sheets.

Samples	Zn	Mg	Cu	Fe	Cr	Mn	Ti	Al
LF	6.34	2.43	1.56	0.13	0.05	0.05	0.09	Bal.
SF	6.23	2.58	1.53	0.12	0.06	0.04	0.08	
NF	6.19	2.53	1.52	0.03	0.02	0.05	0.10	

**Table 2 materials-17-05924-t002:** The average grain size (D), YS, UTS, elongation, n, $\bar{n}$, $\bar{r}$, and Δr for recrystallized sheets as the average of three measurements.

Samples	D (μm)	Direction (°)	YS (MPa)	UTS (MPa)	Elongation (%)	*n*	$\bar{n}$	r	$\bar{r}$	Δr
LF	36.9	0	146.4	336.5	24.6	0.262	0.258	0.56	0.69	−0.13
		45	138.9	321.5	25.8	0.253		0.76		
		90	144.4	326.5	24.4	0.259		0.70		
SF	16.5	0	168.6	361.2	28.2	0.303	0.288	0.72	0.76	−0.07
		45	146.1	324.1	28.5	0.278		0.79		
		90	139.6	321.3	27.6	0.283		0.73		
NF	81.9	0	141.5	318.4	35.8	0.199	0.209	0.81	0.83	−0.05
		45	128.3	313.9	34.3	0.206		0.86		
		90	120.6	309.6	30.3	0.222		0.79		

## Data Availability

The raw/processed data required to reproduce these findings cannot be shared at this time due to technical or time limitations.

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
