# Peer review of "Effect of Al7Cu2Fe Particles on the Anisotropic Mechanical Properties and Formability of Al-Zn-Mg-Cu-Based Alloy Sheets"

_materials, 2024, doi:10.3390/ma17235924_

Round 1
Reviewer 1 Report
Comments and Suggestions for Authors
Here are some observations and suggestions for improving the abstract of your scientific article:
Abstract
· The abstract is relatively dense with information. Consider breaking down complex sentences into simpler ones to enhance readability. For instance, the sentence starting with "As the particles were finely dispersed..." could be simplified for clarity.
· While the abstract provides specific findings, it could benefit from a brief introduction to the significance of studying Al7Cu2Fe particles in Al-Zn-Mg-Cu alloys. A one-sentence context on why this research is relevant would help frame the study better.
· The abstract mentions the preparation of sheets with different particle sizes but does not briefly describe the methods used for these preparations. Including a concise statement about the experimental methods could provide readers with context on how results were obtained.
· The abstract presents results regarding yield strength and elongation but lacks a discussion on what these results imply in practical terms. A sentence discussing the implications of high yield strength and low elongation, or vice versa, would enhance the understanding of the results' significance.
· The term "elongation" should be consistently referred to as "ductility" if that is the intended meaning. Consider clarifying or using terminology consistently throughout the document.
· While the abstract provides quantitative data (e.g., specific yield strengths and elongations), presenting them in a more comparative way could highlight the differences between the samples more effectively. For example, a summary sentence could encapsulate the performance differences between the SF and NF sheets.
· The discussion of anisotropy is somewhat abrupt. Expanding on why the SF and NF sheets exhibited different anisotropic behaviors would add depth to the findings. It might be useful to define anisotropy in the context of your study for clarity.
Introduction
· The introduction could benefit from a clearer structure. Consider organizing the content into distinct paragraphs that introduce the problem, discuss the current state of research, highlight knowledge gaps, and then outline the objectives of your study. This structure can help guide the reader through your argument.
· The introduction starts with the importance of lightweight Al alloys in the automotive and aerospace industries, but it would be helpful to elaborate on the specific benefits these materials bring beyond energy savings and environmental protection. Adding more context on how these alloys contribute to overall vehicle performance could enhance relevance.
· When discussing the limited use of Al-Zn-Mg-Cu alloys in automotive applications, a brief comparison with 5xxx and 6xxx alloys regarding specific properties (like cost, weight, and performance) could help readers understand the relative advantages and disadvantages.
· The discussion on the detrimental effects of coarse intermetallic (FI) particles on mechanical properties mentions several aspects (ductility, fatigue, etc.), but it could be improved by clearly linking these effects to real-world applications. Consider providing specific examples of failure modes in applications to underscore the importance of your research.
· The introduction identifies a gap in research on the modification of Al7Cu2Fe particles in Al-Zn-Mg-Cu alloys but could more explicitly state the objectives of your study. A clear statement of what the study aims to accomplish (e.g., to improve formability and mechanical properties through specific methods) would provide a more focused purpose.
· While the last paragraph outlines the approach taken in this study, it would be beneficial to briefly mention the significance of the chosen methods (e.g., cooling rates, thermo-mechanical treatment) and how they relate to addressing the identified issues with FI particles.
· The introduction ends somewhat abruptly without summarizing the significance of the findings or the broader implications of the research. A concluding statement that ties the introduction together, reiterating the importance of understanding FI particle behavior in improving Al-Zn-Mg-Cu alloys, would provide a stronger conclusion.
Experimental Procedures
· In some instances, temperatures are presented without specifying units. For instance, when mentioning the melting temperature and heat treatment temperatures, ensure that Celsius (°C) is clearly stated for clarity.
· The section could benefit from more detail regarding the specific procedures used for melting and casting. For example, specifying the atmosphere in the furnace (e.g., inert, oxidizing) can provide insight into the processing conditions.
· The cooling rate determination using K-type thermocouples is mentioned, but it would be helpful to explain how these rates were calculated. Providing details on the method or formula used could enhance transparency.
· While the chemical composition analysis method (optical emission spectrometry) is cited, including more details on how the measurements were performed, the accuracy, and any potential limitations would strengthen the methodology section.
· The cold rolling and recrystallization parameters are listed, but a rationale for choosing these specific conditions (temperature, time, reduction rates) could help justify your experimental design. Additionally, elaborating on the implications of these parameters on the resulting microstructure and properties would enhance the reader's understanding.
· The tensile testing conditions are described, but including more details such as the gauge length of the specimens and the method of aligning them relative to the rolling direction could be useful.
· It would be beneficial to mention the statistical methods used to analyze the data, especially since averages and standard deviations are reported. This would provide context on the reliability and significance of the results.
Experimental Results
· The section can benefit from clearer subheadings to delineate distinct parts of the results. For example, sections could be divided into microstructural observations, mechanical properties, and deformation analysis for better readability.
· When discussing results such as average grain sizes and mechanical properties, briefly interpreting these findings in terms of how they relate to the alloy's performance would add depth. For example, explaining why smaller grain sizes or specific textures correlate with improved mechanical properties can enhance reader understanding.
· The results mention anisotropic properties but do not elaborate on why this anisotropy occurs or its implications in practical applications. A brief discussion could clarify the significance of these findings in terms of material selection for specific applications.
· When reporting on the mechanical properties, consider providing a comparative analysis or contextualization with previous studies. This could demonstrate how your findings fit into the broader field of research on Al-Zn-Mg-Cu alloys.
· Ensure that all technical terms, such as "Guinier-Preston zones," are defined or explained when they first appear to make the document accessible to readers who may not be familiar with these concepts.
Discussion
· While the section effectively discusses particle-stimulated nucleation (PSN), further elaboration on the implications of this phenomenon for practical applications (e.g., how it can influence manufacturing processes) would add value.
· The text states that the PSN effect induced by Al7Cu2Fe particles is stronger than other eutectic particles. It would be beneficial to provide a comparison or specific examples to illustrate this point more clearly.
· While the section effectively discusses the yield strength (YS) and the Hall-Petch relationship, including a statement on the practical significance of these findings (e.g., how they could impact material selection for engineering applications) would strengthen the discussion.
· When discussing calculated values, it would enhance clarity to briefly explain how these calculations were performed or the significance of the results derived from the equations presented.
· The text discusses the role of Al7Cu2Fe particles in micro-crack formation. Consider linking this observation to potential improvements in material design, addressing how controlling particle characteristics could enhance mechanical performance.
· The explanation regarding elongation in the NF sheet could be clearer. A more direct comparison with the other sheets, particularly discussing how ductility is related to particle presence and grain size, would be beneficial.
· While the discussion of anisotropic properties is informative, a deeper analysis of how these properties impact real-world applications (e.g., automotive or aerospace components) would provide context and relevance to the findings.
· When discussing the factors influencing anisotropic properties, consider integrating insights or findings from previous studies. This could help reinforce your arguments and provide a more comprehensive view of the topic.
Conclusion
· The conclusion mentions several concepts that have already been discussed in the results section. Summarizing the findings without repeating them verbatim can help streamline the text. For example, instead of reiterating the specific values for yield strength (YS) and elongation, summarize their significance and impact.
· Point 3 provides a good analysis of anisotropy, but it could benefit from more detail on why these findings matter in practical terms. Discussing how anisotropic properties might affect the application of these alloys in industry (e.g., in automotive or aerospace sectors) could enhance relevance.
· The conclusion would be stronger with a final statement that encapsulates the overall contribution of the study. A sentence summarizing the significance of the findings and suggesting potential future work or applications could provide a strong finish.
· While not always included in conclusions, a brief mention of possible future research directions or implications for industrial applications would provide context for readers and highlight the relevance of your findings.
Author Response
<Responses to reviewer>
Dear reviewer,
Thank you very much for your thorough and insightful review of our manuscript. We greatly appreciate the time and effort you have invested in evaluating our work. Your comments have provided valuable guidance, and we believe we have addressed all of the concerns you have raised. Below, we have outlined our responses to each of the points you mentioned, along with the corresponding revisions in the manuscript.
Reviewer 1:
Comments - Abstract
- The abstract is relatively dense with information. Consider breaking down complex sentences into simpler ones to enhance readability. For instance, the sentence starting with "As the particles were finely dispersed..." could be simplified for clarity.
- While the abstract provides specific findings, it could benefit from a brief introduction to the significance of studying Al7Cu2Fe particles in Al-Zn-Mg-Cu alloys. A one-sentence context on why this research is relevant would help frame the study better.
- The abstract mentions the preparation of sheets with different particle sizes but does not briefly describe the methods used for these preparations. Including a concise statement about the experimental methods could provide readers with context on how results were obtained.
- The abstract presents results regarding yield strength and elongation but lacks a discussion on what these results imply in practical terms. A sentence discussing the implications of high yield strength and low elongation, or vice versa, would enhance the understanding of the results' significance.
- The term "elongation" should be consistently referred to as "ductility" if that is the intended meaning. Consider clarifying or using terminology consistently throughout the document.
- While the abstract provides quantitative data (e.g., specific yield strengths and elongations), presenting them in a more comparative way could highlight the differences between the samples more effectively. For example, a summary sentence could encapsulate the performance differences between the SF and NF sheets.
- The discussion of anisotropy is somewhat abrupt. Expanding on why the SF and NF sheets exhibited different anisotropic behaviors would add depth to the findings. It might be useful to define anisotropy in the context of your study for clarity.
Author’s response: Thank you for your valuable comment. According to your suggestion, we have carefully revised the abstract as follows:
“The presence of Al7Cu2Fe particles, formed due to Fe impurities in Al-Zn-Mg-Cu alloys, signifi-cantly impacts mechanical properties and formability, which is crucial for the use of these alloys in automotive and aerospace industries. In this study, we prepared Al-Zn-Mg-Cu-based alloy sheets with large (LF), small (SF), and no (NF) Al7Cu2Fe particles to explore their effects on recrystalli-zation and mechanical behavior. These sheets were fabricated through controlled cooling rates and subsequent thermo-mechanical processing. Fine dispersion of Al7Cu2Fe particles in the SF sheet led to a substantial reduction in grain size (16.5 μm) and an increase in yield strength (168.6 MPa), albeit with lower ductility (24.6%). In contrast, the NF sheet exhibited a lower yield strength (141.5 MPa) but superior ductility (35.8%) due to the absence of Al7Cu2Fe particles, which pre-vented premature fracture. The SF sheet demonstrated lower anisotropy in ductility due to the uniform recrystallized grain orientations, while the LF and NF sheets exhibited significant ani-sotropy in yield strength. These findings indicate that optimizing Al7Cu2Fe particle dispersion is key to balancing strength, ductility, and anisotropy in Al-Zn-Mg-Cu alloys.”
Comments - Introduction
- The introduction could benefit from a clearer structure. Consider organizing the content into distinct paragraphs that introduce the problem, discuss the current state of research, highlight knowledge gaps, and then outline the objectives of your study. This structure can help guide the reader through your argument.
- The introduction starts with the importance of lightweight Al alloys in the automotive and aerospace industries, but it would be helpful to elaborate on the specific benefits these materials bring beyond energy savings and environmental protection. Adding more context on how these alloys contribute to overall vehicle performance could enhance relevance.
- When discussing the limited use of Al-Zn-Mg-Cu alloys in automotive applications, a brief comparison with 5xxx and 6xxx alloys regarding specific properties (like cost, weight, and performance) could help readers understand the relative advantages and disadvantages.
- The discussion on the detrimental effects of coarse intermetallic (FI) particles on mechanical properties mentions several aspects (ductility, fatigue, etc.), but it could be improved by clearly linking these effects to real-world applications. Consider providing specific examples of failure modes in applications to underscore the importance of your research.
- The introduction identifies a gap in research on the modification of Al7Cu2Fe particles in Al-Zn-Mg-Cu alloys but could more explicitly state the objectives of your study. A clear statement of what the study aims to accomplish (e.g., to improve formability and mechanical properties through specific methods) would provide a more focused purpose.
- While the last paragraph outlines the approach taken in this study, it would be beneficial to briefly mention the significance of the chosen methods (e.g., cooling rates, thermo-mechanical treatment) and how they relate to addressing the identified issues with FI particles.
- The introduction ends somewhat abruptly without summarizing the significance of the findings or the broader implications of the research. A concluding statement that ties the introduction together, reiterating the importance of understanding FI particle behavior in improving Al-Zn-Mg-Cu alloys, would provide a stronger conclusion.
Author’s response: Your suggestion to revise the introduction part is mainly related to the practical influence and clearer purpose of this study that we absolutely agree. According to your feedback, we have completely revised the introduction part to clearly present the main novelty and the practical impact of our research as follows:
“Lightweight Al alloys are increasingly regarded as essential materials in the automotive and aerospace industries due to their ability to improve energy efficiency and reduce environmental impact [1, 2]. In addition to these benefits, Al alloys offer a high strength-to-weight ratio, which enhances vehicle performance by contributing to fuel savings, increased payload capacity, and improved handling. Among the widely used Al alloys, the 5xxx (Al-Mg) and 6xxx (Al-Si-Mg) series are prevalent in automotive sheet applications, thanks to their excellent formability and corrosion resistance. However, the 7xxx (Al-Zn-Mg-Cu) alloys, while offering superior stiffness and specific strength, have seen limited use in automotive applications, primarily due to their poor formability. This limits their use to parts like front and rear pillars, and side impact beams, where high strength is prioritized over ease of manufacturing [3]. A comparison between these alloys reveals that while 7xxx alloys provide higher strength, they are costlier and more difficult to process than 5xxx and 6xxx alloys, which are more versatile for forming complex shapes.
The challenge with Al-Zn-Mg-Cu alloys lies in their mechanical limitations, notably their poor formability, which restricts their broader application in automobiles. These alloys are predominantly used in the aerospace sector, where the material can be utilized as thick plates without the need for extensive forming. Therefore, addressing these mechanical deficiencies is crucial for expanding the use of 7xxx alloys in the automotive industry. Improving both their mechanical properties and formability is essential to realize their full potential as a lightweight alternative to steel in critical automotive components.
Research over the last several decades has focused on the formation behavior of Fe-containing intermetallic (FI) particles and their adverse effects on the mechanical properties of Al alloys. During solidification, coarse FI particles like Al3Fe, Al3Fe4, β-AlFeSi, and Al7Cu2Fe form due to the low solubility of Fe in solid Al, even though Fe has high solubility in liquid Al [4]. These particles remain undissolved after homogenization at high temperatures above 500 °C [5-7]. In Al-Zn-Mg-Cu alloys, FI particles larger than 1 μm negatively impact ductility, fatigue resistance, fracture toughness, and corrosion resistance [8-14]. For instance, in automotive crash simulations or high-stress environments, these particles lead to failure modes like void formation and crack propagation, as the brittle FI particles easily fracture, initiating voids that grow and eventually cause material failure [8]. This is particularly relevant in parts subjected to dynamic loads, where the volume fraction of voids increases significantly with rising Fe content, changing the fracture behavior [9]. For example, fracture toughness in Al-Zn-Mg-Cu alloys decreases from 40 to 30 MPa⋅m1/2 as Fe content increases from 0 to 0.6 wt.% [10]. Additionally, Al7Cu2Fe particles provide preferential sites for fatigue crack initiation and promote pitting corrosion, both of which are critical issues in the automotive and aerospace sectors [11-14].
Addressing the detrimental effects of FI particles on mechanical properties requires controlling their size, morphology, and distribution. This can be achieved through alloying element addition, cooling rate control, and innovative processing technologies. For example, the introduction of Mn, Sr, and La has been shown to modify the size and shape of β-AlFeSi particles in Al-Si-Mg alloys [15-17], while rapid cooling rates can reduce the formation of Al3Fe and β-AlFeSi particles [18-21]. Khalifa et al. [22] demonstrated that ultrasonic melt treatment could convert long, plate-like FI particles into fine, polyhedral particles, dramatically improving their mechanical effects. However, limited research has been conducted on the modification of Al7Cu2Fe particles in Al-Zn-Mg-Cu alloys through cooling rate control, a gap this study seeks to address.
This study aims to investigate the effects of FI particles on the mechanical properties and formability of Al-Zn-Mg-Cu alloys. By preparing alloys with varying FI particle content—large, small, and absent—and controlling cooling rates (8.2 and 0.55 K/s), we aim to enhance our understanding of how FI particles affect anisotropic mechanical behavior. Microstructural evolution during thermo-mechanical processing, deformation, and fracture will be closely examined to provide insights into improving the performance of these alloys for automotive applications.”
Comments – Experimental Procedures, Experimental Results, Discussion, Conclusion
- The observations and suggestions for improving the Experimental Procedures, Experimental Results, Discussion, Conclusion of your scientific article.
Author’s response: We have thoroughly reviewed the paper once again and carefully considered your comments and suggestions. As a result, we have revised the Experimental Procedures, Experimental Results, Discussion, and Conclusion sections in the manuscript.
We sincerely hope that the revisions we have made satisfactorily address your concerns and improve the clarity and quality of our manuscript. Should you have any further questions or require additional clarifications, we would be more than happy to provide them.
Once again, thank you for your constructive feedback, and we look forward to your response.
Sincerely,
Heon Kang
Customized Manufacturing R&D Department
Korea Institute of Industrial Technology, Siheung 15014, Republic of Korea
heonkang@kitech.re.kr

Reviewer 2 Report
Comments and Suggestions for Authors
Jeon et al. prepared Al-Zn-Mg-Cu alloy sheets with Al7Cu2Fe intermetallic compounds (FIs) of different sizes and contents. They observed impacts of the Al7Cu2Fe particles on the recrystallization of the Al grains and consequently, on the mechanical performances of the cast samples. The experimental processed were described in detail. The samples were also characterized well. The obtained information might be of use for people working in the related fields. The text falls in the core category of this Journal. The manuscript is written properly. Thus, I’d like to suggest acceptance of this manuscript for publication in Materials after minor improvements.
1). It might better if the authors use the chemical formula in a systematic way. For example, Al13Fe4 and Al3Fe (line 44) have been often used by different authors to present the same compound θ-Al13Fe4.
2. All the references should be in a unified style. There are typos in the References, e.g. Ref. 6 (Line 460).
Author Response
<Responses to reviewer>
Dear reviewer,
Thank you very much for your thorough and insightful review of our manuscript. We greatly appreciate the time and effort you have invested in evaluating our work. Your comments have provided valuable guidance, and we believe we have addressed all of the concerns you have raised. Below, we have outlined our responses to each of the points you mentioned, along with the corresponding revisions in the manuscript.
Reviewer 2:
Comments: Jeon et al. prepared Al-Zn-Mg-Cu alloy sheets with Al7Cu2Fe intermetallic compounds (FIs) of different sizes and contents. They observed impacts of the Al7Cu2Fe particles on the recrystallization of the Al grains and consequently, on the mechanical performances of the cast samples. The experimental processed were described in detail. The samples were also characterized well. The obtained information might be of use for people working in the related fields. The text falls in the core category of this Journal. The manuscript is written properly. Thus, I’d like to suggest acceptance of this manuscript for publication in Materials after minor improvements.
1). It might better if the authors use the chemical formula in a systematic way. For example, Al13Fe4 and Al3Fe (line 44) have been often used by different authors to present the same compound θ-Al13Fe4.
Author’s response: Thank you for your valuable comment. According to your suggestion, we have thoroughly reviewed the paper once again and carefully considered your comments and suggestions. We have revised Al13Fe4 and Al3Fe to θ-Al13Fe4 in the manuscript.
Comments: 2) All the references should be in a unified style. There are typos in the References, e.g. Ref. 6 (Line 460).
Author’s response: We have unified the reference style in accordance with your comments.
We sincerely hope that the revisions we have made satisfactorily address your concerns and improve the clarity and quality of our manuscript. Should you have any further questions or require additional clarifications, we would be more than happy to provide them.
Once again, thank you for your constructive feedback, and we look forward to your response.
Sincerely,
Heon Kang
Customized Manufacturing R&D Department
Korea Institute of Industrial Technology, Siheung 15014, Republic of Korea
heonkang@kitech.re.kr

Reviewer 3 Report
Comments and Suggestions for Authors
1. The contrast and brightness of EDS maps in Fig 2 should be adjusted. Now they are too dark to read.
2. What is the accuracy and noise floor level for the optical emission spectrometry equipment? Especially when it reports some elements of 0.0x wt%, how reliable the number is?
3. In Fig. 3, it is recommended to add SEM images of the samples after recrystallization.
4. I feel confused when comparing Fig. 3 and Fig. 5. The TEM images has almost three orders of magnitude higher magnification than the SEM images here, indicating the intermetallic particle becomes near three order of magnitude smaller after the recrystallization?
If they are the same phase, how could the recrystallization step make the secondary phase particles smaller?
If they are different phase, what happened to the large particles after crystallization? And are the smaller particles already there in the previous steps, just too small to be seen under SEM?
5. It is unfair to compare the samples with such huge difference in grain size. It is very normal and very common that the same material with smaller grains shows better strength but lower ductility. To study the influence of the amount, distribution and morphology of the secondary phase particles, it is important to ensure the grain size of the FCC matrix stay consistent.
Indeed, the FI particles would hinder the grain growth, but it is still straightforward to adjusting the crystallization time to control the grain size (i.e., shortening the crystallization time for the NF sample)
6. To determine the texture of the material via EBSD, there should be a good amount of grains in the field of view. However, it is obviously insufficient for the NF samples.
Comments on the Quality of English LanguageNo comment
Author Response
<Responses to reviewers>
Dear reviewer,
Thank you very much for your thorough and insightful review of our manuscript. We greatly appreciate the time and effort you have invested in evaluating our work. Your comments have provided valuable guidance, and we believe we have addressed all of the concerns you have raised. Below, we have outlined our responses to each of the points you mentioned, along with the corresponding revisions in the manuscript.
Reviewer 3:
Comments #1: The contrast and brightness of EDS maps in Fig 2 should be adjusted. Now they are too dark to read.
Author’s response: Thank you for your valuable comment. I have adjusted the contrast and brightness of the EDS maps in Figure 2 as suggested. The images should now be clearer and easier to read. We have replaced the modified figure in the manuscript.
Comments #2: What is the accuracy and noise floor level for the optical emission spectrometry equipment? Especially when it reports some elements of 0.0x wt%, how reliable the number is?
Author’s response: According to the specification of the OES equipment, the accuracy is <0.01 wt.%, and this is the reason that Table 1 marked the 2nd decimal place. Each alloy composition was measured more than 5 times, and the presented number is the averaged value, allowing reliability in a manner.
Comments #3 & 4: In Fig. 3, it is recommended to add SEM images of the samples after recrystallization. I feel confused when comparing Fig. 3 and Fig. 5. The TEM images has almost three orders of magnitude higher magnification than the SEM images here, indicating the intermetallic particle becomes near three order of magnitude smaller after the recrystallization? If they are the same phase, how could the recrystallization step make the secondary phase particles smaller? If they are different phase, what happened to the large particles after crystallization? And are the smaller particles already there in the previous steps, just too small to be seen under SEM?
Author’s response: Micro-scale particles are Al7Cu2Fe, Al3Mg3CuZn2 that remain even after homogenization heat treatment (Figure 3). Nanoscale particles are particles containing transition metal elements such as Cr, Mn, and Fe, and are different from micro-scale particles (Figure 5).
Micro-scale particles serve as heterogeneous nucleation sites for recrystallized grains by the PSN during the recrystallization (Fig. 16). The micro-scale particles are remain after recrystallization (Fig. 8(a)-(c)).
The nano-scale particles exist even before the recrystallization process, but they cannot be observed by SEM images because of their small size (approximately 45-55 nm). When Al alloys contain small particles, the recrystallization behavior changes significantly. Finely spaced small particles (<500 nm) inhibit the migration of grain boundaries (known as Zener pinning effect). Recrystallization is delayed or prevented by the small particles, and then the Al alloys show a large grain size.
Comments #5: It is unfair to compare the samples with such huge difference in grain size. It is very normal and very common that the same material with smaller grains shows better strength but lower ductility. To study the influence of the amount, distribution and morphology of the secondary phase particles, it is important to ensure the grain size of the FCC matrix stay consistent.
Indeed, the Fl particles would hinder the grain growth, but it is still straightforward to adjusting the crystallization time to control the grain size (i.e., shortening the crystallization time for the NF sample)
Author’s response: We appreciate your detailed comment and understand your concerns regarding the comparison of samples with different grain sizes. However, we would like to clarify that our primary objective in this study is to investigate the influence of secondary phase particles—specifically, their amount, distribution, and morphology—on the mechanical properties of Al-Zn-Mg-Cu alloys. Although grain size is indeed a significant factor in material strength and ductility, the differences in grain size between our samples are an intrinsic outcome of varying the cooling rates and the presence of FI particles.
As you mentioned, FI particles serve as a pinning point of grain growth, enabling proper control of grain size. However, maintaining a uniform grain size across samples with different FI particle distributions would require artificially altering the recrystallization process. Doing so would not reflect the natural metallurgical phenomena during processing and practicality. Therefore, attempting to EQUALIZE the grain sizes would obscure the natural relationships between FI particles and the Al matrix.
Moreover, the strength increase due to grain refinement is not critically significant when compared to the total strength. According to the well-known Hall-Petch relationship (), the constant for Al-Zn-Mg alloy is k = 120 MPa⋅μm0.5 [R1], which is relatively small considering the typical grain size of Al alloys. As shown in Fig. 12, the contribution of grain size reduction to strengthening is approximately 16.7 MPa, which is only a fraction of the total strength of Alloy D (506.7 MPa). Furthermore, the increase in strength due to grain refinement does not always result in a decrease in ductility. In fact, grain boundaries can act as nucleation and sink sites for dislocations [R2], accommodating more dislocations, or complicating the crack propagation path [R3], which can enhance fracture resistance and, in turn, improve ductility.
We hope this clarifies our approach and justifies the chosen methodology.
[R1] J.G. Jeon, K.M. Choi, S. Lee, H. Kang, J.W. Lee, M.R. Joo, D.H. Bae, Mater. Sci. Eng., A, 861 (2022) 144316
[R2] M. Grabski, R. Korski, Phil. Mag. 22 (1970) 707-715
[R3] T. Kobayashi, Mater. Sci. Eng., A 280 (2000) 8-16
Comments #5: To determine the texture of the material via EBSD, there should be a good amount of grains in the field of view. However, it is obviously insufficient for the NF samples.
Author’s response: Regarding the NF sample's EBSD images. In response to your comment, we have taken additional EBSD images to ensure the reliability of the data. Among these, we selected a representative image for inclusion in the manuscript.
We believe that the grains shown in the EBSD image for the NF sample are sufficient to characterize the texture of the material. Furthermore, in the case of coarser grains, substantial grain growth has occurred, and most of the grains are oriented along the basal slip planes. Therefore, the selected EBSD images adequately represent the overall texture and provide a clear understanding of the grain orientations in the NF sample.
We sincerely hope that the revisions we have made satisfactorily address your concerns and improve the clarity and quality of our manuscript. Should you have any further questions or require additional clarifications, we would be more than happy to provide them.
Once again, thank you for your constructive feedback, and we look forward to your response.
Sincerely,
Heon Kang
Customized Manufacturing R&D Department
Korea Institute of Industrial Technology, Siheung 15014, Republic of Korea
heonkang@kitech.re.kr

Reviewer 4 Report
Comments and Suggestions for Authors
The manuscript presents in a well-structured manner a large number of results from the study of the effect of Al7Cu2Fe particles on the anisotropic mechanical properties and formability of Al-Zn-Mg-Cu-based alloy sheets, using many modern analytical methods. The results are strongly based on the differences between Al-Zn-Mg-Cu alloys with different contents of Fe-containing intermetallic particles - large, small and absent and understanding how they affect the anisotropic mechanical behavior.
There are still some things that need to be improved and perfected before being accepted for publication:
1. In the introduction section some concerning aspects related to Fe-intermetallic particles are mentioned, namely in Lines 55-60: “In Al-Zn-Mg-Cu alloys, FI particles larger than 1 μm negatively impact ductility, fatigue resistance, fracture toughness, and corrosion resistance [8- 14]. For instance, in automotive crash simulations or high-stress environments, these particles lead to failure modes like void formation and crack propagation, as the brittle FI particles easily fracture, initiating voids that grow and eventually cause material failure [8].”
And also in Lines 63-65: “Al7Cu2Fe particles provide preferential sites for fatigue crack initiation and promote pitting corrosion, both of which are critical issues in the automotive and aerospace sectors [11-14].”
Could you please explain the choice made for the study of the effect of Al7Cu2Fe particles on the anisotropic mechanical properties and formability of Al-Zn-Mg-Cu based alloy sheets, considering that the literature shows that these critical issues (pitting corrosion and fatigue cracking) related to Al7Cu2Fe particles are problematic for the automotive and aerospace sectors? The question would be whether under these conditions it is still of interest to study these alloys?
2. The references are mainly old; those published in the last 5 years account for about 25-26% of all the references. The proportion of references for the last 5 years should be 1/3 of the total.
3. The colors are not distinguishable in Figure 2, as described in the text in Lines 127-128: "In the EDS analysis, Al, Zn, Mg, Cu, and Fe are indicated by blue, green, red, light green, and orange colors, respectively. " I suggest choosing other colors to differentiate chemical elements.
4. The text in line 136: "The SEM images of as-cast alloys shown in Figure 3(a)-(c)." what was intended to show? The first impression is that it does not contain a verb and needs correction/completion/deletion.
5. Figure 11 is unclear. I suggest choosing other colors to highlight the elements Zn,Mg,Cu, Fe.
6. In the conclusion section in lines 458-462 it is mentioned “This study contributes to a deeper understanding of how particle size and distribution affect the mechanical and anisotropic properties of Al-Zn-Mg-Cu alloys, with direct implications for the automotive and aerospace industries. By refining the control of Al7Cu2Fe particles, manufacturers can develop materials with enhanced strength, ductility, and formability, while minimizing anisotropy.”
My suggestion to the authors is to mention what would be the estimated production cost to bring these Al-Zn-Mg-Cu alloys to market by refining Al7Cu2Fe particle control and whether it is a cost-effective choice for the current market economy.
Author Response
<Responses to reviewer>
Dear reviewer,
Thank you very much for your thorough and insightful review of our manuscript. We greatly appreciate the time and effort you have invested in evaluating our work. Your comments have provided valuable guidance, and we believe we have addressed all of the concerns you have raised. Below, we have outlined our responses to each of the points you mentioned, along with the corresponding revisions in the manuscript.
Reviewer 4:
Comment 1: Could you please explain the choice made for the study of the effect of Al7Cu2Fe particles on the anisotropic mechanical properties and formability of Al-Zn-Mg-Cu based alloy sheets, considering that the literature shows that these critical issues (pitting corrosion and fatigue cracking) related to Al7Cu2Fe particles are problematic for the automotive and aerospace sectors? The question would be whether under these conditions it is still of interest to study these alloys?
Author’s response: Research is needed to improve the mechanical properties of Al-Zn-Mg-Cu alloys by controlling the size, morphology, and distribution of FI particles, as the formation of these particles cannot be fully suppressed. Due to the high solubility of Fe in liquid Al and its low solubility in solid Al, Fe atoms that exceed the solid solubility limit inevitably form FI particles. Furthermore, FI particles remain undissolved even after high-temperature homogenization. However, few studies have investigated the effects of modified FI particles on the microstructure and mechanical properties of Al-Zn-Mg-Cu alloys. This study aims to examine how FI particles influence the mechanical properties and formability of Al-Zn-Mg-Cu alloys by preparing alloys with varying FI particle content (large, small, and absent) and controlling cooling rates (8.2 and 0.55 K/s). The findings from this study will assist manufacturers in developing materials with enhanced strength, ductility, and formability while minimizing anisotropy through the modification of FI particles.
Comment 2: The references are mainly old; those published in the last 5 years account for about 25-26% of all the references. The proportion of references for the last 5 years should be 1/3 of the total.
Author’s response: Thank you for your comment. According to your feedback, we have added several up-to-date references as follows:
[3] M.K. Gupta, V. Singhal, Review on materials for making lightweight vehicles, Mater. Today Proc. 56(2) (2022) 868-872.
[5] S.–S. Li, X. Yue, Q.–Y. Li , H.–L. Peng, B–X. Dong, T.–S. Liu, H.–Y. Yang, J. Fan, S.–L. Shu, F. Qiu, Q.–C. Jiang, Development and applications of aluminum alloys for aerospace industry, J. Mater. Res. Technol. 27 (2023) 944-983.
[7] B. Wang , X. Liu , J. Wang, Q. Li , K. Liu, M. Zhang, Uncovering the effects of Ce and superheat temperature on Fe-rich intermetallic and microporosity formation in aluminum alloy, Mater. Charact. 193 (2022) 112226.
[8] T. Liu, M. Karkkainen, L. Nastac, V. Arvikar, I. Levin, L.N. Brewer, Iron-rich intermetallics in high pressure die cast A383 aluminum alloys, Intermetallics 126 (2020) 106814.
[13] I. Goto, K. Shirai, R. Ohyama, K. Kurosawa, Dissolution Mechanism of Intermetallic Layer by Iron Erosion in Aluminum-Based Molten Binary Alloys, Mater. Trans. 63 (2022) 730-739.
[61] B.B. Wang, G.M. Xie, L.H. Wu, P. Xue, D.R. Ni, B.L. Xiao, Y.D. Liu, Z.Y. Ma, Grain size effect on tensile deformation behaviors of pure aluminum, Mater. Sci. Eng., A 820 (2021) 141504.
[63] W. Chae, M. Jeong, D. Lee, J. Lee, D.W. Chun, S.Y. Lee, S.-K. Hong, S.H. Kim, J.H. Han, Effects of pre/post-aging treatment on the mechanical properties and texture of asymmetrically rolled 6061 aluminum alloy: formability and planar anisotropy, J. Mater. Res. Technol. 24 (2023) 9476-9490.
[64] A. Dhal, S.K. Panigrahi, M.S. Shunmugam, A comprehensive study on size-effect, plastic anisotropy and microformability of aluminum with varied alloy chemistry, crystallographic texture, and microstructure, Mater. Sci. Eng., A 876 (2023) 145111.
Comment 3: The colors are not distinguishable in Figure 2, as described in the text in Lines 127-128: "In the EDS analysis, Al, Zn, Mg, Cu, and Fe are indicated by blue, green, red, light green, and orange colors, respectively. " I suggest choosing other colors to differentiate chemical elements.
Author’s response: We have adjusted the brightness and contrast of Figure 2 to make the colors representing the chemical elements more distinguishable. The revised figure has been replaced in the manuscript to ensure clearer differentiation of the elements as described in the text.
Comment 4: The text in line 136: "The SEM images of as-cast alloys shown in Figure 3(a)-(c)." what was intended to show? The first impression is that it does not contain a verb and needs correction/completion/deletion.
Author’s response: Thank you for pointing this out. We have deleted the text in line 136 to improve clarity and coherence in the manuscript.
Comment 5: Figure 11 is unclear. I suggest choosing other colors to highlight the elements Zn,Mg,Cu, Fe.
Author’s response: We have adjusted the brightness and contrast of Figure 11 to make the colors representing the chemical elements more distinguishable. The revised figure has been replaced in the manuscript.
Comment 6: In the conclusion section in lines 458-462 it is mentioned “This study contributes to a deeper understanding of how particle size and distribution affect the mechanical and anisotropic properties of Al-Zn-Mg-Cu alloys, with direct implications for the automotive and aerospace industries. By refining the control of Al7Cu2Fe particles, manufacturers can develop materials with enhanced strength, ductility, and formability, while minimizing anisotropy.”.
My suggestion to the authors is to mention what would be the estimated production cost to bring these Al-Zn-Mg-Cu alloys to market by refining Al7Cu2Fe particle control and whether it is a cost-effective choice for the current market economy
Author’s response: The grain refinement technology through Al7Cu2Fe phase control is significantly influenced by the solidification rate and has been designed with the strip-casting method. This method cost-effective casting process) allows the desired phase size to be achieved due to the rapid solidification process, eliminating the need for additional processes for phase control. The sheets produced in this way exhibit excellent formability due to the control of anisotropy, offering a potential solution to overcome the current forming limitations in the aluminum components industry.
We sincerely hope that the revisions we have made satisfactorily address your concerns and improve the clarity and quality of our manuscript. Should you have any further questions or require additional clarifications, we would be more than happy to provide them.
Once again, thank you for your constructive feedback, and we look forward to your response.
Sincerely,
Heon Kang
Customized Manufacturing R&D Department
Korea Institute of Industrial Technology, Siheung 15014, Republic of Korea
heonkang@kitech.re.kr

Round 2
Reviewer 1 Report
Comments and Suggestions for Authors
Dear Authors,
I have carefully reviewed the revised version of your manuscript titled "Effect of Al7Cu2Fe Particles on the Anisotropic Mechanical Properties and Formability of Al-Zn-Mg-Cu Alloy Sheets." I am pleased to inform you that the revisions have addressed the reviewers' comments adequately, and I now consider the manuscript suitable for publication.
Author Response
<Responses to reviewer>
Dear reviewer,
Thank you very much for your thorough and insightful review of our manuscript. We greatly appreciate the time and effort you have invested in evaluating our work. Your comments have provided valuable guidance, and we believe we have addressed all of the concerns you have raised. Below, we have outlined our responses to each of the points you mentioned, along with the corresponding revisions in the manuscript.
Reviewer 1:
Comments: I have carefully reviewed the revised version of your manuscript titled "Effect of Al7Cu2Fe Particles on the Anisotropic Mechanical Properties and Formability of Al-Zn-Mg-Cu Alloy Sheets." I am pleased to inform you that the revisions have addressed the reviewers' comments adequately, and I now consider the manuscript suitable for publication.
Author’s response: Thank you very much for your thorough review and for considering our revised manuscript, "Effect of Al7Cu2Fe Particles on the Anisotropic Mechanical Properties and Formability of Al-Zn-Mg-Cu Alloy Sheets," suitable for publication. We deeply appreciate the constructive feedback provided throughout the review process, which has significantly improved the quality of our work. In addition, we have conducted a thorough English language editing to enhance the clarity and readability of the manuscript.
We sincerely hope that the revisions we have made satisfactorily address your concerns and improve the clarity and quality of our manuscript. Should you have any further questions or require additional clarifications, we would be more than happy to provide them.
Once again, thank you for your constructive feedback, and we look forward to your response.
Sincerely,
Heon Kang
Customized Manufacturing R&D Department
Korea Institute of Industrial Technology, Siheung 15014, Republic of Korea
heonkang@kitech.re.kr

Reviewer 3 Report
Comments and Suggestions for Authors
I am unconvinced by authors' explanation for not controlling the grain size. It is crucial, and basic, to ensure that the matrix FCC phase has similar grain size, to study the influence of the secondary phase. It is not hard, though indeed requires extra experiment.
Besides, in response to comment #6, authors claim that they “have taken additional EBSD images to ensure the reliability of the data”. However, they didn’t provide any more images or data to support this statement.
The response to comment #3 and #4 makes sense. It is recommended to add it to the paper.
Comments on the Quality of English LanguageNo comment
Author Response
<Responses to reviewer>
Dear reviewer,
Thank you very much for your thorough and insightful review of our manuscript. We greatly appreciate the time and effort you have invested in evaluating our work. Your comments have provided valuable guidance, and we believe we have addressed all of the concerns you have raised. Below, we have outlined our responses to each of the points you mentioned, along with the corresponding revisions in the manuscript.
Reviewer 3:
Comment 1: I am unconvinced by authors' explanation for not controlling the grain size. It is crucial, and basic, to ensure that the matrix FCC phase has similar grain size, to study the influence of the secondary phase. It is not hard, though indeed requires extra experiment.
Author’s response: Thank you for your feedback. We agree that it is ideal to compare the influence of the secondary phase within a matrix that has a consistent grain size. However, due to differences in solidification rates, achieving identical grain sizes across samples is challenging. Nonetheless, we will refer to relevant studies in the literature to support the findings and provide further context for this aspect of the study.
Comment 2: Besides, in response to comment #6, authors claim that they “have taken additional EBSD images to ensure the reliability of the data”. However, they didn’t provide any more images or data to support this statement.
Author’s response: Thank you for your comment. To support our response to comment #6, we have included an additional low-magnification EBSD image in Figure 4(c) to ensure the reliability of the data. This image complements the existing data and provides a clearer representation of the grain structure.
Figure 4. IPF maps and ODFs of the (a, d) LF, (b, e) SF, and (c, f) NF sheets recrystallized at 480 ℃ for 30 m. Low-magnification IPF map of the NF sheet in Figure 4(c).
Comment 3: The response to comment #3 and #4 makes sense. It is recommended to add it to the paper.
Author’s response: We have incorporated the responses to comments #3 and #4 into the paper as suggested.
We sincerely hope that the revisions we have made satisfactorily address your concerns and improve the clarity and quality of our manuscript. Should you have any further questions or require additional clarifications, we would be more than happy to provide them.
Once again, thank you for your constructive feedback, and we look forward to your response.
Sincerely,
Heon Kang
Customized Manufacturing R&D Department
Korea Institute of Industrial Technology, Siheung 15014, Republic of Korea
heonkang@kitech.re.kr
